# ROSE: Reordered SparseGPT for More Accurate One-Shot Large Language Model Pruning

## Abstract

Pruning is widely recognized as an effective method for reducing the parameters of large language models (LLMs), potentially leading to more efficient inference. One classic and prominent path of one-shot LLM pruning is to leverage the second-order gradients (*i.e.*, Hessian), represented by the pioneering works like SparseGPT (Frantar & Alistarh, 2023). However, the predefined left-to-right pruning order in SparseGPT leads to suboptimal performance when the weights exhibit *columnar* patterns. This paper studies the effect of pruning order under the SparseGPT framework. The analyses lead us to propose ROSE, a reordered SparseGPT method that prioritizes weight columns with larger potential pruning errors to be processed first. Specifically, following the block-wise iterative pruning scheme of SparseGPT, we first perform a pre-pruning step to identify weights that are highly likely to be pruned, based on which we compute both column-wise and block-wise pruning loss. Columns within each block are then reordered in descending order of column loss, while blocks are reordered in descending order of block loss. We further analyze different layer types and selectively apply reordering to specific layers. Substantial empirical results on prevalent LLMs (LLaMA2-7B/13B/70B, LLaMA3-8B, Mistral-7B) demonstrate that ROSE surpasses the original SparseGPT and other counterpart pruning methods.

## 1 Introduction

Large language models (LLMs) (Thoppilan et al., 2022; Achiam et al., 2023; Bai et al., 2023; Guo et al., 2025; Yang et al., 2024; Zhang et al., 2022) have demonstrated remarkable capabilities in natural language understanding and generation attributed to the massive scale of their architectures and training data, achieving significant breakthroughs across a wide range of natural language processing tasks such as machine translation (Zhang et al., 2025), text summarization (Wang et al., 2025), and question answering (Brown et al., 2025). However, with hundreds of billions of parameters, these models require substantial memory and computational resources, posing significant challenges for deployment on resource-constrained devices (Steiner et al., 2023).

Model pruning is an effective way to enhance model inference and deployment efficiency by removing less critical weights or neurons while maintaining competitive performance. Traditional pruning methods typically determine which weights to prune in a single pass based on designed criteria (Molchanov et al., 2019; Yang et al., 2023; Li et al., 2017), or iteratively select the weights with the smallest pruning error for removal, followed by retraining the remaining weights to recover performance (Liu et al., 2017; Han et al., 2015; Zhang et al., 2023; Hoefler et al., 2022).

Nevertheless, retraining-based approaches become prohibitively expensive and time-consuming for LLMs given the significant computational cost required for full-model optimization. On the contrary, pruning approaches for LLMs focus on post-training pruning (PTP) methods. Following the classic paradigms of traditional pruning, some current approaches directly prune weights in a single pass but without further adjusting the remaining parameters. Wanda (Sun et al., 2023) takes outliers (Dettmers et al., 2022) in LLM and proposes a new pruning metric combining weight magnitude and input activations. DSnoT (Zhang et al., 2024) introduces a retraining-free approach by adjusting the mask after initial pruning. In the domain of iterative pruning, SparseGPT (Frantar & Alistarh, 2023) employs layer-wise compensation based on pioneering works OBS (Hassibi & Stork, 1993) and OBC (Frantar & Alistarh, 2022). It adopts a fixed left-to-right order to prune individual columns and

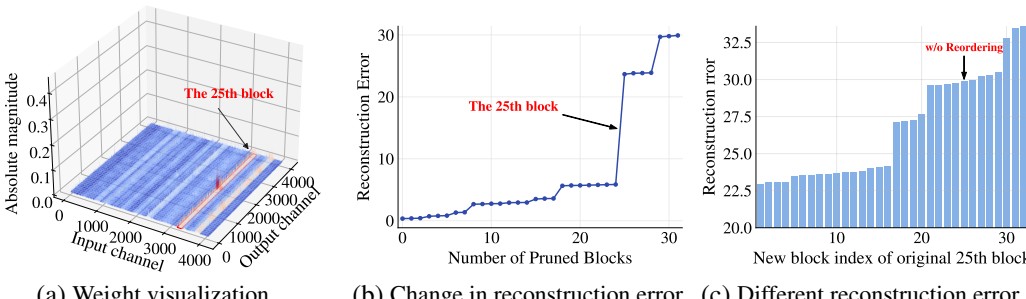

(a) Weight visualization    (b) Change in reconstruction error  (c) Different reconstruction error

Figure 1: (a) Weight visualization of the "self_attn.o_proj" layer in the LLaMA2-7B layer0. It exhibits a *columnar* pattern along the input channels, and the 25th block contains a concentration of weights with large magnitudes. (b) Change in reconstruction error during SparseGPT pruning as the number of pruned blocks increases. There is a sharp increase in reconstruction error when pruning the 25th block. (c) Different reconstruction error after reordering the original 25th block while maintaining the relative positions of other blocks. The earlier the original 25th block is pruned, the smaller the reconstruction error.

uses the weights from columns that have not yet been pruned for compensation. However, weights with larger pruning losses are pruned in later stages, and the smaller remaining parameter set may be insufficient to compensate for the greater loss.

In large language models, we observe that the weights of certain layers (e.g., the output projection layer of self-attention) exhibit a distinct *columnar* distribution and weights with large magnitudes are clustered, as illustrated in Figure 1(a). SparseGPT employs a fixed-size *iterative blocking* strategy to generate pruning masks, which can introduce large reconstruction errors, particularly for blocks[1] with highly concentrated large weights. As shown in Figure 1(b), a sharp increase in reconstruction error occurs when pruning the 25th block, which is located within the region where high-magnitude weights are concentrated, as illustrated in Figure 1(a). Since this block is located toward the latter part of the matrix, very few remaining weights are available at this stage to compensate for the large pruning error.

To further investigate this phenomenon, we move this block from the leftmost to the rightmost end while preserving the relative order of all other blocks and apply the SparseGPT framework to prune. The reconstruction error of different pruning orders is shown in Figure 1(c). The earlier this block is pruned, the smaller the final reconstruction error and vice versa. This observation leads to the following question: *Can we achieve a better weight reconstruction in SparseGPT by proposing an optimized pruning order?*

In this paper, we introduce `ROSE`, a one-shot pruning order adjustment method based on SparseGPT, to examine the above question. We first perform a pre-pruning step to identify weights that are highly likely to be pruned, based on which we compute both column-wise and block-wise pruning losses. Then columns within each block are then reordered in descending order of their losses, while blocks themselves are reordered according to their block losses. We further analyze different types of layers and selectively apply reordering to specific layers. Experimental results demonstrate that `ROSE` can surpass original SparseGPT and other existing unstructured pruning methods on prevalent LLMs. Extensive evaluations on representative LLMs show the efficacy of our proposed approach. Our contributions are as follows:

- We find that a key factor in accurate one-shot pruning based on the SparseGPT framework is the pruning order, and propose `ROSE` to study the problem for the first time.
- We propose a method to determine a more optimal pruning order in a single step. Specifically, we first perform a pre-pruning to select candidate less important weights, based on which we compute column importance scores and block importance scores. The column order within each block is then reordered according to column scores, and the block order is rearranged according to block scores.
- Empirically, extensive evaluations on prevalent models suggest our method performs favorably against prior SoTA counterparts.

---

[1]Here, the term "block" refers to the sub-matrix along input channels within the weight matrix.

## 2 RELATED WORK

### 2.1 NETWORK PRUNING

Network pruning aims to reduce redundant parameters while maintaining model accuracy (LeCun et al., 1990; Hassibi & Stork, 1993). In terms of workflow, traditional methods can be broadly divided into two categories: *global pruning*, which first determines "what to prune" based on a certain importance criterion and then retrains the remaining weights to recover performance (Li et al., 2017; Molchanov et al., 2019; Yang et al., 2023). and *iterative pruning*, which involves repeated cycles of pruning based on certain criteria, subsequent fine-tuning of the remaining weights, and re-assessment of the pruning criteria. Methods in this category generally adopt a greedy order, pruning weights in ascending order of pruning error (Singh & Alistarh, 2024; Liu et al., 2017; Han et al., 2015). Although these methods have achieved remarkable success in balancing sparsity and accuracy, they all follow a fixed pruning paradigm. To date, no work has explored the impact of pruning order on final model performance.

### 2.2 UNSTRUCTURED PRUNING FOR LLMS

Unstructured pruning aims to remove unimportant individual weights from the network (LeCun et al., 1990; Han et al., 2016), which differs from structured pruning that aims to remove entire structures such as channels, attention heads, filters, and layers (Michel et al., 2019; Li et al., 2017; Fang et al., 2023; He et al., 2017). Unstructured pruning preserves the original model structure and can be performed in a training-free manner (Frantar & Alistarh, 2022), an advantage that is particularly critical under conditions of constrained fine-tuning resources in the era of LLMs. SparseGPT (Frantar & Alistarh, 2023) pioneers one-shot unstructured pruning for LLMs via layer-wise Hessian-based reconstruction. It can prune models containing hundreds of billions of parameters with a single GPU, without fine-tuning and maintaining negligible accuracy. Recent years have witnessed growing attention toward unstructured methods for large language models. For example, Wanda (Sun et al., 2023) proposes a simple yet effective metric combining weight magnitudes and input activation norms. DSnoT (Zhang et al., 2024) dynamically grows and adjusts the pruning mask to reduce the reconstruction error. OATS (Zhang & Papyan, 2025) approximates each weight matrix as the sum of a sparse matrix and a low-rank matrix, while preserving the critical outlier features of the LLMs.

## 3 PREREQUISITES

**Layer-Wise Pruning.** Layer-wise pruning aims to optimize the structure of neural networks by selectively removing less significant weights within each layer. The pruning process for a given layer $l$ is formulated as a minimization problem of the $\ell_2$-error between the original and pruned outputs, defined as follows:

$$\text{argmin}_{\hat{\mathbf{W}}_l} ||\mathbf{W}_l \mathbf{X}_l - \hat{\mathbf{W}}_l \mathbf{X}_l||_2^2, \tag{1}$$

where $\mathbf{W}_l$ represents the weight matrix before pruning, $\hat{\mathbf{W}}_l$ is the pruned weight matrix, and $\mathbf{X}_l$ denotes the input to the layer $l$.

**Optimal Brain Surgeon (OBS) Framework.** Optimal Brain Surgeon (OBS) is based on a Taylor expansion of the loss function. Removes the weight with minimal impact on the objective function and updates the remaining weights to minimize the change in loss. Given a weight matrix $\mathbf{W} \in \mathbb{R}^{M \times N}$ and the corresponding input data matrix $\mathbf{X} \in \mathbb{R}^{D \times N}$, let $\mathbf{H} = \mathbf{X}\mathbf{X}^\mathsf{T}$ denote the corresponding Hessian matrix, the increase in loss $\Delta\mathcal{L}$ caused by the removal of the weight $w_q$ and the optimal updating of the remaining weights $\Delta\mathbf{w}$ are given by:

$$\Delta\mathcal{L} = \frac{w_q^2}{[\mathbf{H}^{-1}]_{qq}}, \quad \Delta\mathbf{w} = -\frac{w_q}{[\mathbf{H}^{-1}]_{qq}} \mathbf{H}_{:,q}^{-1}. \tag{2}$$

In the process of pruning, OBS employs an iterative update approach that involves multiple Hessian inverse operations, leading to high computational complexity for large-scale models. To address these problems, OBC (Frantar & Alistarh, 2022) decomposes the objective of the layer-wise reconstruction into subproblems by row and proposes an efficient computational framework for matrix inversion

through optimized Gaussian elimination. The inverse of $\mathbf{H}$ with its $p$-th row and column removed $\mathbf{H}_{-p}^{-1}$ can be calculated via $\mathbf{H}^{-1}$:

$$\mathbf{H}_{-p}^{-1} = \left( \mathbf{H}^{-1} - \frac{1}{[\mathbf{H}^{-1}]_{pp}} \mathbf{H}_{:,p}^{-1} \mathbf{H}_{p,:}^{-1} \right)_{-p}, \tag{3}$$

where $\mathbf{H}_{:,p}^{-1}$ and $\mathbf{H}_{p,:}^{-1}$, represent the $p$-th column and row of $\mathbf{H}$, respectively, and the $(\cdot)_{-p}$ operator indicates the subsequent removal of the $p$-th row and column.

**Revisiting SparseGPT.** SparseGPT (Frantar & Alistarh, 2023) is a one-shot pruning method designed specifically for LLMs. It decouples pruning into mask selection and iterative weight reconstruction based on the OBS framework and leverages the remaining unupdated weights for compensation to achieve pruning acceleration. For mask selection, SparseGPT selects the pruning mask for $B_S$ columns at a time and adaptively chooses the mask while running the reconstruction. For weight reconstruction, it employs a left-to-right pruning order and the first row of the intermediate inverse matrices obtained by successively applying Equation 3 is stored in each row of the upper triangular matrix from the Cholesky decomposition of $\mathbf{H}^{-1}$ (Frantar et al., 2022). That is,

$$(\mathbf{H}_{i:,i:})_{1,:}^{-1} = \mathbf{L}_{i,i} \cdot (\mathbf{L}^{\mathsf{T}})_{i,:}, \tag{4}$$

where $\mathbf{H}^{-1} = \mathbf{L}\mathbf{L}^{\mathsf{T}}$ and $\mathbf{L}$ is upper triangular matrix.

## 4 METHODOLOGY

In this section, we first analyze the dilemma of adjusting the pruning order within the SparseGPT framework and give our main solutions. Then, we propose ROSE: (1) By performing a pre-pruning step, we select out weights with an extremely high probability of being pruned to estimate pruning loss. (2) By reordering the columns within one block and blocks in descending order of pruning loss, the columns with greater potential errors are pruned earlier.

### 4.1 ANALYSES

In order to prioritize pruning columns with greater pruning errors earlier, we first need to obtain the pruning loss of each column. Following the existing method, column prune loss is defined as the sum of the single loss of pruned weights in that column. Then, to calculate this loss, we need to know in advance which weights in each column will be pruned—that is, the pruning mask. In the SparseGPT pruning process, after pruning one block, the mask for the next block is determined based on the updated weights. This means we cannot obtain the complete mask information before pruning begins, so we cannot obtain the loss of each column until the true pruning process. Therefore, the pruning order should be determined iteratively. Yet, the Cholesky decomposition implicitly stores the cumulative information of all weight updates throughout the entire pruning process, which means that the pruning order must be determined in advance.

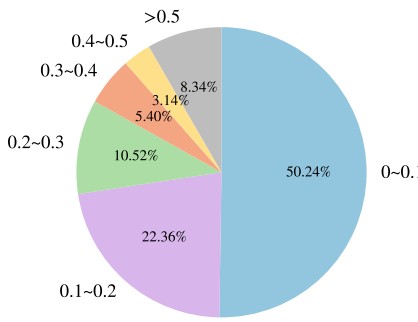

Figure 2: The distribution of relative change of weights before and after pruning. The majority of weights do not change a lot compared with their original magnitudes.

Given such a contradiction, we address this problem from a different perspective. Our observation is that for the vast majority of weights, the post-pruning magnitudes deviate only marginally from their original values, as shown in Figure 2. This suggests that the relative importance of most weights remains largely stable during the pruning process. Therefore, we can follow the iterative blocking of SparseGPT to identify weights that are highly likely to be pruned before pruning, and use their initial pruning loss as an approximation of pruning loss in the subsequent process, based on which we can reorder columns to prioritize those with higher estimated error to be pruned first. Furthermore, mask selection is performed within each block. In order to allow the mask selection to continue functioning within its original local context, we should ensure that the weights within a block remain grouped together. Therefore, the entire block should be treated as a unit for reordering.

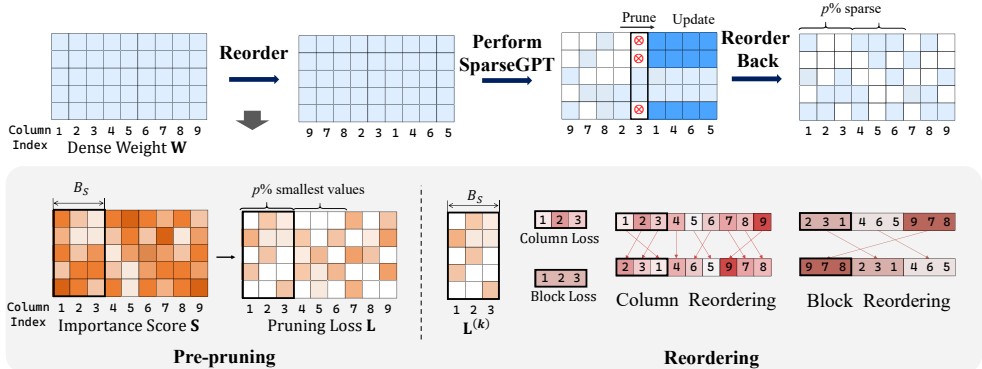

Figure 3: Illustration of ROSE (Darker colors indicate higher importance scores). Given the dense weight $\mathbf{W}$ and input activation matrix $\mathbf{X}$, we calculate the importance score $\mathbf{S}$ and split it into blocks based on $B_S$. Each block $\mathbf{S}^{(k)}$ has a sparsity of $p\%$ to form the pruning loss sub-matrix $\mathbf{L}^{(k)}$, which is used to obtain column scores and block scores. Within each block, the columns are reordered in descending order by column scores, and the blocks are reordered in descending order by block scores. Then, pruning is performed with SparseGPT. The obtained sparse weight matrix is reordered back to the original order.

### 4.2 ROSE

**Pre-pruning.** We first need to compute the importance score for each weight. We adopt the metric proposed in Wanda (Sun et al., 2023), which combines the weight magnitude and the corresponding input activation. Specifically, given the weight matrix $\mathbf{W} \in \mathbb{R}^{M \times N}$, where $M$ donates the number of output channels and $N$ donates the number of input channels, let $\mathbf{X} \in \mathbb{R}^{(B \times L) \times N}$ represent the input activation matrix, where $B$ represents the batch size and $L$ denotes the sequence length. For a single weight $\mathbf{W}_{ij}$, the importance score $\mathbf{S}_{ij}$ is defined as:

$$\mathbf{S}_{ij} = |\mathbf{W}_{ij}| \cdot \|\mathbf{X}_j\|_2, \tag{5}$$

where $\cdot$ represents the element-wise product and $\|\mathbf{X}_j\|_2$ denotes the $\ell_2$ norm of the corresponding input activation.

Next, we perform a pre-pruning step to select candidate weights that are highly likely to be pruned during the actual pruning process. We follow the block-wise iterative framework of SparseGPT. Specifically, $\mathbf{W}$ is divided into blocks $K$ where $K = \lceil N/B_s \rceil$ along the dimension of the column. The corresponding weight sub-matrix can be denoted as $\mathbf{W}^{(k)} = \mathbf{W}[:, i_1 : i_2]$ where $i_1 = (k-1) \cdot B_s$ and $i_2 = \min(k \cdot B_s, N)$. The corresponding input activation sub-matrix can be denoted as $\mathbf{X}^{(k)} = \mathbf{X}[:, i_1 : i_2]$. Then we can get score sub-matrix $\mathbf{S}^{(k)}$ by Equation 5. Let $p\%$ be the target sparsity rate. For each block $k = 1, 2, \ldots, K$, the smallest $p\%$ of elements in $\mathbf{S}^{(k)}$ are extracted to form $\mathbf{L}^{(k)}$, which corresponds to the importance scores of the weights for pre-pruning.

**Reordering.** Our reordering process includes two levels: column reordering and block reordering. For column reordering, we sort them in descending order according to their respective column pruning loss within each block. Specifically, the pruning loss of each column in block $k$ is:

$$l_j^{(k)} = \sum_{i=1}^{M} \left[ \mathbf{L}^{(k)} \right]_{ij}, \quad j = i_1, i_1 + 1 \ldots, i_2. \tag{6}$$

Subsequently, the columns of $\mathbf{W}^{(k)}$ are sorted in descending order based on $l_j^{(k)}$. For block reordering, we treat the entire block as a single unit and perform a descending order rearrangement. The total block loss is calculated as follows:

$$L^{(k)} = \sum_{i=1}^{M} \sum_{j=i_1}^{i_2} \left[ \mathbf{L}^{(k)} \right]_{ij}, \quad k = 1, \ldots, K. \tag{7}$$

The blocks $\mathbf{W}^{(1)}, \ldots, \mathbf{W}^{(K)}$ are reordered in descending order based on the values of $L^{(k)}$.

## 5 EXPERIMENTAL RESULTS

### 5.1 EXPERIMENT SETTINGS

**Models and Datasets**. We select the current public large language model (LLM) for evaluation, including the LLaMA2 series (Touvron et al., 2023), LLaMA3 series (Dubey et al., 2024), and Misrtal-7B (Jiang et al., 2023). These models range in size from 7 billion to 70 billion parameters. We primarily assess the performance of pruned large language models through perplexity, a widely adopted and stable metric for measuring LLM performance, and use WikiText-2-raw (Merity et al., 2016) datasets. To further evaluate the capabilities of pruned models, we also conduct experiments on seven standard common-sense benchmark tasks: BoolQ (Clark et al., 2019), WinoGrande (Sakaguchi et al., 2021), PIQA (Bisk et al., 2020), OpenBookQA (Mihaylov et al., 2018), HellaSwag (Zellers et al., 2019), ARC-Easy and ARC-Challenge (Clark et al., 2018). All zero-shot tasks are uniformly performed based on the lm-eval-harness framework (Sutawika et al., 2013).

**Comparison Methods.** We compare our method with counterpart methods, including: (1) Magnitude-based (Han et al., 2016) pruning that removes weights based on the magnitude metric. (2) SparseGPT (Frantar & Alistarh, 2023) that utilizes approximate second-order Hessian information to evaluate weight importance and perform weight reconstruction. (3) Wanda (Sun et al., 2023) that removes weights based on magnitudes multiplied by the corresponding input activation norms. (4) DSnoT (Zhang et al., 2024) that performs training-free fine-tuning with dynamic masks after pruning. (5) OATS (Zhang & Papyan, 2025) that decomposes weight matrices into a sparse matrix and a low-rank matrix, with a designed strategy to preserve critical outlier features.

**Implementation Details.** `ROSE` is implemented with the PyTorch framework (Paszke et al., 2019) and HuggingFace Transformers (Wolf et al., 2019). Consistent with previous works (Frantar & Alistarh, 2023), the calibration data contains 128 samples randomly selected from the first shard of the C4 dataset (Raffel et al., 2020). Each sample contains sequences of 2048 tokens. All experiments are conducted on NVIDIA 24GB 4090 GPUs or NVIDIA 48GB 4090 GPUs. For the same model, we ensure that all methods are run on the same GPUs. The results of comparison methods are reproduced by its official code. DSnoT is combined with both Magnitude Wanda and SparseGPT in its paper. In our experiments, we run them both and take the best one to put in the results. For SparseGPT and `ROSE` , the blocksize is set to 128.

### 5.2 RECONSTRUCTION ERROR ANALYSES

In this section, we evaluate the reconstruction error of the single layer using different methods.

First, we perform a comparison between `ROSE` and SparseGPT. We try to apply `ROSE` to all layers of the LLaMA2-7B model at 70% sparsity rate and use the ratio of the reconstruction error to the original SparseGPT reconstruction error as an evaluation metric. Figure 4 shows the mean and deviation across all different types of layers. `ROSE` consistently reduces the reconstruction error across all O matrices. The most significant reduction achieves 35%. The changes in reconstruction error for other types of matrices are not significant. This result is consistent with the visualization of the weight distribution. Except for the O matrices, the weights of the other matrices are overall relatively uniformly distributed along the column dimension, namely, the

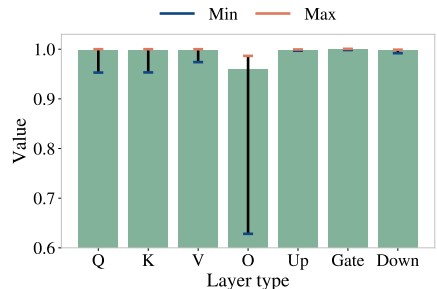

Figure 4: Mean of the reconstruction error ratio (our method vs. SparseGPT) across layers in the LLaMA2-7B model at 70% sparsity. A ratio less than 1 indicates our method achieves a smaller reconstruction error.

input channel. Since SparseGPT uses block-wise masking with the same sparsity in each block, the pruning error differences between different blocks are small. Whether reordering is performed has little impact on the reconstruction error. In contrast, the weight distribution of the O matrix is highly uneven, and some blocks introduce significant errors during pruning, making reordering effective in reducing its reconstruction error.

Then, we analyze the reconstruction error by all methods. The results are shown in Table 1. It can be observed that Wanda, DSnoT, and Magnitude, which prune weights directly, exhibit a significant

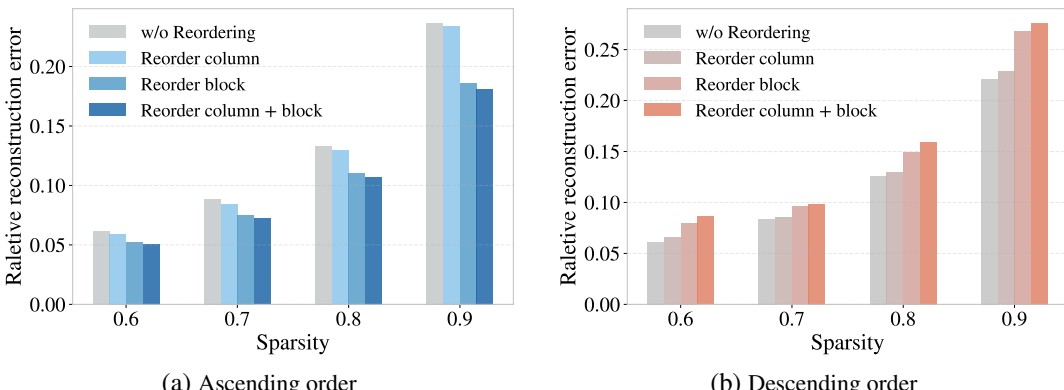

(a) Ascending order                  (b) Descending order

Figure 5: Relative reconstruction error of the O matrix in LLaMA2-7B layer1 by ROSE and variants at varying sparsity rates.

Table 1: Relative reconstruction error of the O matrix in LLaMA2-7B layer1 by different methods.

| Sparsity | Magnitude | Wanda | SparseGPT | DSnoT | OATS (ours) | ROSE |
|---|---|---|---|---|---|---|
| 60% | 1.50e-1 | 9.39e-2 | 6.12e-2 | 9.39e-1 | 1.50e-1 | **5.09e-2** |
| 70% | 2.20e-1 | 1.40e-1 | 8.84e-2 | 1.40e-1 | 2.00e-1 | **7.22e-2** |
| 80% | 3.20e-1 | 2.20e-1 | 1.33e-1 | 2.20e-1 | 3.00e-1 | **1.07e-1** |
| 90% | 5.00e-1 | 3.60e-1 | 2.37e-1 | 3.60e-1 | 4.80e-1 | **1.81e-1** |

increase in reconstruction error as sparsity increases. OATS, despite decomposing the weights into the sum of a sparse matrix and a low-rank matrix, also shows large reconstruction errors at high sparsity levels. In contrast, SparseGPT and ROSE consistently achieve lower reconstruction errors across different sparsity levels. This is because both methods are based on the OBS second-order framework, allowing effective compensation after weight pruning.

Figure 5 provides a detailed analysis demonstrating that ROSE achieves a lower reconstruction error than SparseGPT. As shown in Figure 5(a), both column reordering and block reordering individually contribute to reducing the reconstruction error, with block reordering yielding more pronounced improvements. Meanwhile, the reduction in reconstruction error becomes more pronounced as the sparsity level increases. Interestingly, Figure 5(b) shows the opposite trend when the order of our method is reversed, pruning the blocks and columns first with smaller errors. It can be seen that both block and column are pruned starting from the smallest pruning error. The reconstruction error increases compared to the original SparseGPT, and when both follow the order of smallest to largest, the reconstruction error increases even further, which is the complete opposite of the descending order. This contrast strongly indicates that the pruning order accounts for pruning error.

### 5.3 MAIN BENCHMARK RESULTS

**Unstructured Pruning Results.** Table 2 presents the WikiText perplexity performance on WikiText and zero-shot task performance of unstructured pruning methods on LLaMA2 models at 70% sparsity rate. In terms of zero-shot evaluations, ROSE achieves better average accuracy across all evaluated models than original SparseGPT. For task-specific evaluations, ROSE can achieve higher accuracy in the majority of tasks across the models of different sizes. Notably, at the 7B, our approach surpasses SparseGPT by over 1.5% in ARC-c and ARC-e tasks.

Table 3 shows the WikiText perplexity performance on LLaMA3-8B and Mistral-7B at different sparsity rates. At 60% sparsity rates, the performance differences among various methods, except for magnitude, are not significant. Across all sparsity rates, ROSE consistently achieves the lowest perplexity, validating its capability to enhance pruned model performance. For example, at 80% sparsity rate, ROSE reduces perplexity from 203.45 to 172.14. Overall, SparseGPT and ROSE outperform other pruning methods in most cases. Moreover, our method achieves better perplexity results than the original SparseGPT.

Table 2: Performance of WikiText perplexity ($\downarrow$) and zero-shot task accuracy ($\uparrow$) for different unstructured pruning methods on LLaMA2 Models at 70% sparsity rate.

| Model | Method | Perplexity | Zero-shot Accuracy | | | | | | | |
|---|---|---|---|---|---|---|---|---|---|---|
| | | | BoolQ | WinoG. | PIQA | OBQA | HellaS. | ARC-e | ARC-c | Avg. |
| | Dense | 5.47 | 77.68 | 69.14 | 79.05 | 44.20 | 76.01 | 74.54 | 46.33 | 66.71 |
| | Magnitude | 4.98e4 | 37.95 | 49.25 | 51.52 | 28.00 | 26.32 | 27.90 | **26.96** | 35.41 |
| | Wanda | 72.58 | 48.50 | 49.33 | 53.86 | 25.80 | 30.21 | 30.64 | 21.33 | 37.10 |
| LLaMA2-7B | SparseGPT | 27.68 | 63.61 | 58.41 | 62.35 | 29.60 | 40.38 | 40.19 | 23.46 | 45.43 |
| | DSnoT | 60.44 | 62.14 | 55.25 | **63.00** | 30.40 | 39.24 | **44.15** | 25.94 | 45.73 |
| | OATS | 50.44 | 60.46 | 51.38 | 55.11 | 28.20 | 32.32 | 32.45 | 21.59 | 40.22 |
| | ROSE (ours) | **26.38** | **64.04** | **59.19** | 62.84 | **30.60** | **41.35** | 41.71 | 25.26 | **46.43** |
| | Dense | 4.88 | 80.55 | 71.98 | 80.52 | 45.20 | 79.36 | 77.53 | 48.98 | 69.16 |
| | Magnitude | 2.14e2 | 38.65 | 49.49 | 53.10 | 26.60 | 29.51 | 32.11 | 24.49 | 36.28 |
| | Wanda | 46.22 | 62.08 | 50.75 | 57.34 | 28.20 | 31.62 | 35.56 | 21.25 | 40.97 |
| LLaMA2-13B | SparseGPT | 19.78 | **68.17** | 61.88 | 67.85 | 32.20 | 46.70 | 48.11 | **28.67** | 50.51 |
| | DSnoT | 31.21 | 64.86 | 56.20 | 66.92 | **33.60** | 47.17 | 49.45 | 27.22 | 49.35 |
| | OATS | 40.80 | 62.42 | 56.04 | 60.39 | 29.20 | 35.27 | 38.01 | 22.61 | 43.42 |
| | ROSE (ours) | **19.54** | 65.90 | **63.22** | **68.01** | 33.00 | **47.61** | **49.54** | 27.99 | **50.75** |
| | Dense | 3.32 | 83.76 | 77.98 | 82.70 | 48.80 | 83.81 | 81.06 | 57.25 | 73.62 |
| | Magnitude | 423.46 | 39.57 | 57.14 | 67.63 | 35.80 | 57.20 | 54.55 | 34.56 | 49.49 |
| | Wanda | 10.59 | 74.10 | 74.03 | 75.63 | 40.00 | 64.73 | 69.99 | 40.78 | 62.75 |
| LLaMA2-70B | SparseGPT | 9.34 | **80.58** | **75.30** | 77.04 | 41.60 | 69.19 | 70.03 | **43.86** | 65.37 |
| | DSnoT | **8.29** | 79.02 | 73.95 | **77.31** | **42.80** | **71.96** | 69.53 | 42.75 | 65.33 |
| | OATS | 9.97 | 75.60 | 73.23 | 75.83 | 40.70 | 68.13 | 69.49 | 41.38 | 63.48 |
| | ROSE (ours) | 9.29 | 80.18 | 75.14 | 76.44 | 42.60 | 69.40 | **70.79** | 43.77 | **65.47** |

Table 3: WikiText perplexity ($\downarrow$) performance on LLaMA3-8B model and Mistral-7B model at varying sparsity rates.

| Method | LLaMA-3 8B (dense: 6.14) | | | | Mistral-7B (dense: 5.32) | | | |
|---|---|---|---|---|---|---|---|---|
| | 60% | 70% | 80% | 90% | 60% | 70% | 80% | 90% |
| Magnitude | 3.38e5 | 1.62e6 | 8.54e6 | 2.35e6 | 31.42 | 8.88e3 | 1.34e4 | 1.22e5 |
| Wanda | 23.34 | 123.78 | 986.97 | 1.02e4 | 11.11 | 57.31 | 236.17 | 5.16e3 |
| SparseGPT | **15.23** | 40.48 | 203.45 | 1.10e3 | 9.37 | 21.48 | **78.69** | 286.54 |
| DSnoT | 19.66 | 126.99 | 995.57 | 8.38e4 | 9.67 | 30.51 | 1.91e3 | 7.76e3 |
| OATS | 16.34 | 88.93 | 770.54 | 7.95e3 | 10.54 | 35.20 | 261.60 | 6.44e3 |
| ROSE (ours) | 15.50 | **40.29** | **172.14** | **840.10** | **9.30** | **20.86** | 78.96 | **266.88** |

Table 4: WikiText perplexity ($\downarrow$) on LLaMA models with 2:4 pattern.

| Method | 2-7B | 2-13B | 3-8B |
|---|---|---|---|
| SparseGPT | 11.00 | 8.77 | 16.33 |
| ROSE (ours) | **10.73** | **8.60** | **15.84** |

Table 5: WikiText perplexity ($\downarrow$) on LLaMA models with 4:8 pattern.

| Method | 2-7B | 2-13B | 3-8B |
|---|---|---|---|
| SparseGPT | 8.46 | 7.00 | 12.20 |
| ROSE (ours) | **8.30** | **6.96** | **12.00** |

**Semi-structured Pruning Results.** ROSE can be extended to semi-structured pruning by changing the pre-pruning step according to the semi-structured sparsity pattern and adjusting the blocksize parameter accordingly. Specifically, the blocksize is set to 4 for the 2:4 pattern and to 8 for the 4:8 pattern. Table 4 and 5 show the perplexity performance of LLaMA models in two semi-structured patterns. The results show that our approach outperforms SparseGPT in both the 2:4 and 4:8 patterns. For example, under the 2:4 pattern, our method reduces by 0.5 compared to SparseGPT, demonstrating the effectiveness and superiority of ROSE in semi-structured pruning.

Table 6: Ablation study of different reordering processes at 70% sparsity rate.

| Method | Reordering Process | | Perplexity (↓) | Average Accuracy (↑) |
| --- | --- | --- | --- | --- |
| | Column | Block | | |
| SparseGPT | × | × | 27.68 | 45.43 |
| ROSE (ours) | ✓ | × | 26.80 | 45.55 |
| ROSE (ours) | × | ✓ | 27.21 | 46.17 |
| ROSE (ours) | ✓ | ✓ | **26.36** | **46.43** |

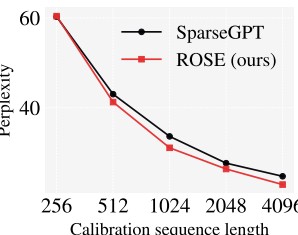

Figure 6: Ablation study of blocksize, calibration samples, and calibration sequence length in LLaMA2-7B at 70% sparsity rates.

## 5.4 ABLATION STUDY

**Reordering Process.** To validate the effectiveness of our two-stage pruning strategy, We separately apply (i) column-only reordering, (ii) block-only reordering, and (iii) both reorderings jointly. Results are shown in Table 6. On the LLaMA2-7B model, both individual column reordering and block reordering improve performance on the WikiText dataset and enhance the zero-shot average accuracy, with their combination yielding further gains. Overall, both reordering strategies demonstrated positive effects.

**Blocksize.** ROSE involves reordering both blocks and the columns within them, a process governed by the blocksize parameter. This section investigates its performance in a controlled comparison with SparseGPT under identical blocksize conditions on LLaMA2-7B at 70% sparsity rate. As illustrated in the Figure 6(a), ROSE exhibits robustness similar to SparseGPT, with WikiText perplexity remaining stable over a wide range of blocksize values and ROSE consistently achieves a lower perplexity than the original SparseGPT.

**Calibration Data.** Since both SparseGPT and ROSE rely on the Hessian matrix computed from calibration data for weight compensation, we analyze the impact of calibration data number and sequence length on the LLaMA2-7B model at 70% sparsity rate. The results are shown in Figure 6(b) and 6(c). For the number of calibration data, ROSE consistently outperforms SparseGPT by achieving lower perplexity. Similarly, longer input sequences also lead to reduced perplexity, and ROSE maintains its performance advantage across all sequence lengths.

## 6 CONCLUSION

This paper introduces ROSE, a new one-shot layerwise pruning method based on the second-order pruning framework. We are motivated by an interesting finding that certain layers in existing LLMs exhibit a *columnar* pattern, and directly applying SparseGPT leads to suboptimal results. We propose ROSE to reorder these layers, allowing columns with higher pruning loss to be processed first, thereby preserving more adjustable parameters. ROSE performs a pre-pruning step to compute both column-wise and block-wise pruning losses, and then reorders columns in descending order of column loss within each block while reordering blocks in descending order of block loss. Extensive results on representative LLMs show the merits of our method against other recent top-performing counterparts.

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

## DECLARATION OF THE USE OF LARGE LANGUAGE MODELS

The use of Large Language Models (LLMs) in this study is strictly limited to grammatical checks and textual refinement. All key contributions, including the conceptual development, framework design, implementation, and experimental analysis, are independently conceived and executed by the authors without the use of LLMs.

## A ROSE ALGORITHM

We present the full procedure of ROSE in Algorithm 1. First, the weight matrix is partitioned into blocks. Within each block, we calculate weight importance scores, perform pre-pruning based on a target sparsity to create a loss matrix, and then reorder the columns by their loss. Blocks are then globally reordered by their total loss, with the activation matrix transformed accordingly and all indices stored. The reordered weight matrix is pruned using SparseGPT, after which the resulting sparse matrix is restored to its original order using the stored indices.

---

**Algorithm 1** The ROSE Algorithm

---

**Input:** Weight matrix $\mathbf{W} \in \mathbb{R}^{M \times N}$, input activation $\mathbf{X} \in \mathbb{R}^{(B \times L) \times N}$, block size $B_s$, target sparsity $p\%$

**Output:** Sparse weight matrix $\mathbf{W}_{\text{sparse}}$

1: **Step 1: Perform Reorder**

2: $K \leftarrow \lceil N/B_s \rceil$

3: $\pi \leftarrow [1, 2, \ldots, K]$

4: **for** $k = 1$ to $K$ **do**

5:     $i_1 \leftarrow (k-1) \cdot B_s$, $i_2 \leftarrow \min(k \cdot B_s, N)$

6:     $\mathbf{W}^{(k)} \leftarrow \mathbf{W}[:, i_1 : i_2]$

7:     $\mathbf{X}^{(k)} \leftarrow \mathbf{X}[:, i_1 : i_2]$        ▷ Follow *iterative blocking* framework

8:     $\mathbf{S}_{ij}^{(k)} \leftarrow |\mathbf{W}_{ij}^{(k)}| \cdot \|\mathbf{X}_j^{(k)}\|_2$        ▷ Calculate importance score of each weight

9:     $\mathbf{L}^{(k)} \leftarrow p\%$ smallest scores in $\mathbf{S}^{(k)}$        ▷ Pre-pruning

10:     $l_j^{(k)} = \sum_{i=1}^M \mathbf{L}_{ij}^{(k)}$ for $j = i_1, \ldots, i_2$        ▷ Calculate column pruning loss

11:     $I^{(k)} \leftarrow \text{argsort}(-l^{(k)})$        ▷ Descending order

12:     $\mathbf{W}^{(k)} \leftarrow \mathbf{W}_{:, \, I^{(k)}}^{(k)}$

13:     $\mathbf{X}^{(k)} \leftarrow \mathbf{X}_{:, \, \text{inner\_perm}_k}^{(k)}$

14:     $L^{(k)} = \sum_{i=1}^M \sum_{j=i_1}^{i_2} \mathbf{L}_{ij}^{(k)}$        ▷ Calculate block pruning loss

15: **end for**

16: $\pi \leftarrow \text{argsort}(-L^{(k)})$

17: $\mathbf{W}_{\text{reorder}} \leftarrow \left[\mathbf{W}^{(\pi(1))}, \ldots, \mathbf{W}^{(\pi(K))}\right]$        ▷ Descending order

18: $\mathbf{X}_{\text{reorder}} \leftarrow \left[\mathbf{X}^{(\pi(1))}, \ldots, \mathbf{X}^{(\pi(K))}\right]$

19: **Step 2: Apply SparseGPT Pruning**

20: $\mathbf{H}_{\text{reorder}} \leftarrow \mathbf{X}_{\text{reorder}} \mathbf{X}_{\text{reorder}}^\top$

21: $\mathbf{W}_{\text{reorder}} \leftarrow \text{SparseGPT}\left(\mathbf{W}_{\text{reorder}}, \mathbf{H}_{\text{reorder}}, p\%, B_S\right)$

22: **Step 3: Reorder Back**

23: $\mathbf{W}_{\text{temp}} \leftarrow \mathbf{W}_{\text{reorder}}[:, \text{argsort}(\pi)]$

24: **for** $k = 1$ to $K$ **do**

25:     $i_1 \leftarrow (k-1) \cdot B_s$, $i_2 \leftarrow \min(k \cdot B_s, N)$

26:     $\mathbf{W}_{\text{sparse}}[:, i_1 : i_2] \leftarrow \mathbf{W}_{\text{temp}}[:, i_1 : i_2]_{:, \, \text{argsort}(I^{(k)})}$

27: **end for**

28: **return** $\mathbf{W}_{\text{sparse}}$

---

# B  MORE EXPERIMENTAL RESULTS

Table 7: Performance of zero-shot task accuracy (↑) for different unstructured pruning methods on LLaMA3-8B Models at different sparsity rates.

| Model | Method | BoolQ | WinoG. | PIQA | OBQA | HellaS. | ARC-e | ARC-c | Avg. |
|---|---|---|---|---|---|---|---|---|---|
| 0% | Dense | 81.38 | 72.61 | 80.79 | 45.00 | 79.17 | 77.74 | 53.50 | 70.03 |
| 70% | Magnitude | 37.80 | 50.12 | 51.74 | 28.40 | 26.40 | 26.64 | 25.09 | 35.17 |
| | Wanda | 55.14 | 47.83 | 54.57 | 26.80 | 29.13 | 30.81 | 21.16 | 37.92 |
| | SparseGPT | **68.78** | 55.96 | 61.04 | **29.60** | 41.11 | 40.11 | **25.26** | 45.98 |
| | DSnoT | 51.47 | 50.75 | 60.99 | 26.80 | 32.90 | 36.15 | 22.53 | 40.23 |
| | OATS | 60.70 | 51.07 | 56.04 | 25.80 | 29.68 | 31.82 | 20.82 | 39.42 |
| | ROSE (ours) | 68.26 | **56.83** | **62.79** | 27.80 | **41.12** | **41.08** | 24.57 | **46.06** |
| 80% | Magnitude | 60.15 | **51.07** | 50.11 | **30.20** | 26.37 | 24.66 | **25.26** | 38.26 |
| | Wanda | 38.96 | 49.64 | 51.69 | 27.20 | 27.96 | 27.27 | 24.06 | 35.25 |
| | SparseGPT | 53.79 | 48.86 | 53.32 | 25.60 | 28.40 | **30.26** | 20.22 | 37.21 |
| | DSnoT | 37.83 | 49.01 | 51.36 | 27.80 | 27.57 | 27.27 | 23.81 | 34.95 |
| | OATS | 37.83 | 49.17 | 52.18 | 27.20 | 27.43 | 27.99 | 24.23 | 35.15 |
| | ROSE (ours) | **61.25** | 50.12 | **53.75** | 27.40 | **28.48** | 29.46 | 21.50 | **38.85** |
| 90% | Magnitude | **62.17** | 50.43 | 52.83 | 29.40 | 26.23 | 25.34 | 26.02 | **38.92** |
| | Wanda | 41.31 | 49.64 | 50.27 | 26.80 | 26.07 | 25.38 | 25.34 | 34.97 |
| | SparseGPT | 38.07 | 49.64 | 51.47 | 25.00 | **27.07** | **28.41** | 23.21 | 34.70 |
| | DSnoT | 45.32 | 49.88 | 48.97 | 25.80 | 25.72 | 25.93 | 24.83 | 35.21 |
| | OATS | 37.83 | 48.93 | 50.60 | 28.40 | 26.35 | 25.67 | **26.88** | 34.95 |
| | ROSE (ours) | 37.83 | 49.88 | 52.23 | 27.00 | 26.97 | 27.53 | 23.46 | 34.99 |

Table 8: Performance of zero-shot task accuracy (↑) for different unstructured pruning methods on Mistral-7B Models at different sparsity rates.

| Model | Method | BoolQ | WinoG. | PIQA | OBQA | HellaS. | ARC-e | ARC-c | Avg. |
|---|---|---|---|---|---|---|---|---|---|
| 0% | Dense | 82.14 | 73.80 | 82.26 | 44.20 | 80.42 | 78.20 | 52.30 | 70.47 |
| 70% | Magnitude | 37.86 | 51.70 | 55.17 | 28.00 | 29.35 | 30.56 | 23.38 | 36.57 |
| | Wanda | 61.77 | 51.78 | 57.40 | 27.00 | 31.75 | 33.12 | 21.67 | 40.64 |
| | SparseGPT | **67.25** | 58.17 | 65.51 | **31.20** | 46.63 | 46.80 | 27.30 | 48.98 |
| | DSnoT | 63.12 | 56.04 | 65.89 | 29.00 | 45.84 | 44.40 | 26.62 | 47.27 |
| | OATS | 60.70 | 51.07 | 56.04 | 25.80 | 29.68 | 31.82 | 20.82 | 39.42 |
| | ROSE (ours) | 65.14 | **60.14** | **66.10** | 30.00 | **46.94** | **48.78** | **27.56** | **49.24** |
| 80% | Magnitude | 49.91 | **51.30** | 50.33 | 25.80 | 25.88 | 26.52 | **26.79** | 36.65 |
| | Wanda | 37.83 | 47.36 | 53.16 | 25.00 | 27.73 | 28.16 | 24.57 | 34.83 |
| | SparseGPT | 53.91 | 50.75 | **54.13** | 26.80 | 29.56 | 29.63 | 20.90 | 37.95 |
| | DSnoT | 37.83 | 46.25 | 52.45 | **27.60** | 27.69 | 29.00 | 22.35 | 34.74 |
| | OATS | 37.83 | 49.17 | 52.18 | 27.20 | 27.43 | 27.99 | 24.23 | 35.15 |
| | ROSE (ours) | **59.76** | 50.20 | 53.86 | 26.60 | **29.60** | **30.05** | 20.82 | **38.70** |
| 90% | Magnitude | **53.30** | **52.72** | 50.98 | 27.40 | 25.99 | 26.22 | 28.24 | **37.84** |
| | Wanda | 37.83 | 50.28 | 50.49 | 27.40 | 26.47 | 27.86 | 27.90 | 35.46 |
| | SparseGPT | 37.83 | 48.93 | **52.29** | 25.80 | 27.06 | 28.37 | 25.09 | 35.05 |
| | DSnoT | 37.83 | 50.43 | 50.22 | 25.40 | 26.29 | 27.57 | **28.50** | 35.18 |
| | OATS | 37.83 | 48.93 | 50.60 | **28.40** | 26.35 | 25.67 | 26.88 | 34.95 |
| | ROSE (ours) | 37.86 | 47.67 | 51.80 | 26.80 | **27.06** | **28.37** | 24.74 | 34.90 |

