# OpenReview forum: "Reordered SparseGPT: Optimizing the Pruning Order in Second-Order LLM Pruning"
_ICLR.cc/2026/Conference — ICLR 2026 Conference Withdrawn Submission_

### Official Review · Reviewer_Qx47 · 2025-10-24

**Soundness:** 2
**Presentation:** 3
**Contribution:** 2
**Rating:** 2
**Confidence:** 5

**Summary:**

This paper presents ROSE (Reordered SparseGPT), a variant of SparseGPT that investigates the impact of pruning order in one-shot pruning for large language models (LLMs). The authors observe that SparseGPT’s fixed left-to-right pruning order can lead to suboptimal results when weight matrices exhibit columnar patterns. ROSE introduces a pre-pruning step to estimate pruning losses and reorders columns and blocks by these losses, ensuring that weights with higher potential errors are pruned earlier. Experiments on LLaMA2, LLaMA3, and Mistral models demonstrate marginal improvements in perplexity and zero-shot accuracy over baseline methods, suggesting that pruning order can modestly enhance pruning efficiency.

**Strengths:**

1. ROSE is an intuitive extension of SparseGPT, making it easy to understand and implement. The motivation is clearly illustrated and empirically supported by targeted analyses (e.g., Figure 1–4).
2. The paper is well-organized, with clear figures, detailed methodology, and comprehensive comparisons across multiple models and sparsity settings.

**Weaknesses:**

1. Limited necessity of the reorder mechanism: As shown in Figure 4, significant improvement occurs mainly in the attn.o_proj layers, while other weights show minimal gains. This suggests that reordering benefits are confined to specific layer structures.
2. Marginal algorithmic novelty: The method only introduces a heuristic reordering atop SparseGPT, without a deeper theoretical or generalizable insight into why this should universally improve pruning.
3. Limited empirical gains: Across Table 2–5, ROSE’s improvements in perplexity and zero-shot accuracy are generally minor (≈1% or less), and often within variance. The method performs roughly on par with SparseGPT.
4. Practical utility concerns: For instance, in Table 2, ROSE’s LLaMA2-70B (70% sparsity) achieves a zero-shot average accuracy of 65.47, still below LLaMA2-13B Dense (69.16). Hence, from a deployment standpoint, the pruning gain does not offset performance degradation, reducing the motivation for such pruning.

**Questions:**

See weaknesses

---

### Official Review · Reviewer_Ehkt · 2025-10-25

**Soundness:** 2
**Presentation:** 2
**Contribution:** 2
**Rating:** 2
**Confidence:** 5

**Summary:**

This paper proposes a LLM pruning method called ROSE (Reordered SparseGPT). The authors observe that the fixed "left-to-right" pruning order employed by SparseGPT leads to suboptimal performance when the weights exhibit columnar patterns. To this end, ROSE introduce a new pruning order optimization mechanism. Specifically, it first identifies weights that are highly likely to be pruned based on the column-wise and block-wise pruning loss. It then reorders the columns and blocks based on these losses. Extensive experiments on multiple LLMs demonstrate that ROSE outperforms SparseGPT and other baseline methods in both PPL and downstream tasks.

**Strengths:**

- The models used in this paper include LLaMA2 (7B, 13B, 70B), LLaMA3-8B, Mistral-7B, covering the current mainstream models.
- This paper is easy to understand.

**Weaknesses:**

- This paper lacks rigorous theoretical proof to prove that the sorting method adopted by ROSE is the "optimal" pruning order.
- As a pruning method, its actual inference acceleration effect on CPU and GPU is not reflected, which makes people doubt its actual acceleration effect.
- The paper only makes a small change based on the existing pruning method SparseGPT, which is not innovative enough.
- There is no comparison between the time and computing cost of pruning and SparseGPT.
- The proposed method has poor generalization and can only be implemented based on SparseGPT.
- As shown in Figure 6b, the calibration samples have a significant impact on the model pruning effect (Perplexity), which proves that the proposed method has poor robustness.

**Questions:**

Compared with the existing channel pruning and layer pruning methods, which one has better hardware acceleration effect?

---

### Official Review · Reviewer_n52K · 2025-10-30

**Soundness:** 2
**Presentation:** 3
**Contribution:** 2
**Rating:** 2
**Confidence:** 3

**Summary:**

The paper proposes **ROSE** (Reordered SparseGPT), a one-shot, second-order pruning method that modifies the pruning order in the SparseGPT framework. Motivated by the observation that some LLM layers, especially the self-attention output projection (O), exhibit columnar weight patterns with clustered large-magnitude columns, the authors argue that the standard left-to-right order in SparseGPT can be suboptimal. ROSE first performs a pre-pruning pass to estimate column-wise and block-wise pruning losses (using a Wanda-style importance score), then reorders columns and blocks in descending loss so that high-loss components are pruned earlier, leaving more unpruned parameters for subsequent OBS-style compensation. Empirically, ROSE reduces layer reconstruction error (most clearly on O matrices) and achieves slightly better perplexity/zero-shot accuracy than baselines across LLaMA2/3 and Mistral at 60–90% sparsity; it also reports small gains when extended to 2:4 and 4:8 semi-structured patterns.

**Strengths:**

- **Clear motivation and tidy algorithm.** Reordering addresses a plausible failure mode of left-to-right pruning when high-loss columns are delayed; the proposed fix is easy to implement within SparseGPT.
- **Consistent (if modest) quality gains.** Across models/datasets, ROSE often reduces reconstruction error and slightly improves perplexity/zero-shot accuracy vs. SparseGPT, with similar benefits observed on semi-structured patterns (2:4 / 4:8).
- **Reproducible setup.** The one-shot calibration protocol, ablation knobs (block size, scoring choice), and baseline alignment make it straightforward for practitioners to try.

**Weaknesses:**

1. **Sparsity ≠ efficiency; missing speedups.** The paper reports sparsity/perplexity/accuracy but no efficiency-related (eg., inference speedup) results on realistic sparse runtimes. The experiments are mainly conducted on unstructured pruning, which often yields limited acceleration without bespoke kernels. Could you provide latency (prefill+decode), tokens/s, and GPU memory vs. dense/SparseGPT under matched decoding and hardware?
2. **Emphasis on very high sparsity with severe accuracy drops.** Many results center on **70–90%** sparsity where accuracy reductions are large (e.g., LLaMA2-7B dense avg ~66.7 vs. pruned ~46.4 at 70%). This is hard to justify for deployment. Please evaluate **0–50%** sparsity and show that ROSE strictly dominates SparseGPT/Wanda/DSnoT in that practical regime.
3. **Incremental improvements over SparseGPT.** The net gains are small, even at extreme sparsity, and sometimes within noise or worse on some tasks.
4. **Pruning-time overhead not reported.** ROSE introduces a pre-scoring + reordering pass. For large models, pruning time and memory overhead matter. Compare end-to-end pruning time and peak memory vs. SparseGPT across model sizes and calibration lengths.

**Questions:**

- Can ROSE yield transferable insights or mechanisms (e.g., scoring, ordering policies) that benefit other compression approaches?
- From a hardware perspective, random (especially unstructured) pruning does not provide a large speed-up because operations like weight matrix multiplication rely on locality and targeting blocks of a matrix at one time, i.e. a model with 50% sparsity is typically much slower than a dense model with 50% of the parameters. Can you demonstrate end-to-end acceleration and/or run-time memory savings for your pruned models versus dense and SparseGPT baselines across multiple sparsity levels?
- How much additional pruning time and peak memory does ROSE incur relative to SparseGPT or other pruning methods?

---

### Official Review · Reviewer_MqrD · 2025-10-31

**Soundness:** 4
**Presentation:** 3
**Contribution:** 3
**Rating:** 6
**Confidence:** 5

**Summary:**

A well-executed and carefully validated improvement to SparseGPT. Although it is a relatively incremental step, the work is meaningful in practice, as it identifies an overlooked design factor that yields consistent, measurable benefits in large-scale model pruning. It also includes additional discussion on generalization and runtime cost.

**Strengths:**

1. Novel Observation on Pruning Order: Identifies a previously overlooked factor, the left-to-right column order in SparseGPT, and empirically demonstrates how it affects reconstruction error, particularly in column-structured layers (e.g., output projection matrices). ROSE integrates smoothly with SparseGPT without retraining or major architectural changes. The approach is conceptually simple yet well-motivated by the underlying mathematical and empirical analysis.
2. Strong and clear empirical validation: experiments across multiple LLMs (LLaMA2-7B/13B/70B, LLaMA3-8B, Mistral-7B) consistently show improvements in perplexity and zero-shot accuracy at various sparsity levels. The paper provides convincing evidence (Figure 1 and Figure 5) linking pruning order to reconstruction error, and ablation studies validate both column- and block-level reordering effects.

**Weaknesses:**

1. The idea, changing pruning order, is conceptually straightforward. While effective, it may be viewed as a small engineering improvement over SparseGPT rather than a fundamentally new framework. The paper relies on empirical intuition (stability of importance scores) rather than a formal theoretical justification for why pre-estimated pruning losses are sufficient.

**Questions:**

1. ROSE appears most effective on O-matrices; other layer types show little benefit. Discussion of how to automatically detect or generalize to other structures could be expanded. Will this choice change when switching to other model architectures?
2. Computational Overhead Not Quantified: While claimed to be lightweight, the added pre-pruning and sorting steps’ runtime overhead is not clearly reported.
3. Recent adaptive pruning strategies [1-2] might provide a stronger baseline for comparison or be discussed in the related work.



[1] Outlier Weighed Layerwise Sparsity (OWL): A Missing Secret Sauce for Pruning LLMs to High Sparsity
[2] AlphaPruning: Using Heavy-Tailed Self Regularization Theory for Improved Layer-wise Pruning of Large Language Models

---

### Note · Authors · 2025-11-13

I have read and agree with the venue's withdrawal policy on behalf of myself and my co-authors.